# Gut Microbial Catabolites of Tryptophan Are Ligands and Agonists of the Aryl Hydrocarbon Receptor: A Detailed Characterization

**DOI:** 10.3390/ijms21072614

**Published:** 2020-04-09

**Authors:** Barbora Vyhlídalová, Kristýna Krasulová, Petra Pečinková, Adéla Marcalíková, Radim Vrzal, Lenka Zemánková, Jan Vančo, Zdeněk Trávníček, Jan Vondráček, Martina Karasová, Sridhar Mani, Zdeněk Dvořák

**Affiliations:** 1Department of Cell Biology and Genetics, Faculty of Science, Palacky University, Slechtitelu 27, 783 71 Olomouc, Czech Republic; vyhlidalovabara@gmail.com (B.V.); kkrasulova@seznam.cz (K.K.); petra.pecinkova@upol.cz (P.P.); Marcalikova.Adela@seznam.cz (A.M.); radim.vrzal@email.cz (R.V.); lenka.zemankova@upol.cz (L.Z.); 2Division of Biologically Active Complexes and Molecular Magnets, Regional Centre of Advanced Technologies and Materials, Faculty of Science, Palacký University, Šlechtitelů 27, 783 71 Olomouc, Czech Republic; jan.vanco@upol.cz (J.V.); zdenek.travnicek@upol.cz (Z.T.); 3Department of Cytokinetics, Institute of Biophysics of the Czech Academy of Sciences, Královopolská 135, 61265 Brno, Czech Republic; vondracek@ibp.cz (J.V.); karasova@ibp.cz (M.K.); 4Department of Genetics and Department of Medicine, Albert Einstein College of Medicine, Bronx, NY 10461, USA

**Keywords:** aryl hydrocarbon receptor, tryptophan, indoles, microbiome

## Abstract

We examined the effects of gut microbial catabolites of tryptophan on the aryl hydrocarbon receptor (AhR). Using a reporter gene assay, we show that all studied catabolites are low-potency agonists of human AhR. The efficacy of catabolites differed substantially, comprising agonists with no or low (i3-propionate, i3-acetate, i3-lactate, i3-aldehyde), medium (i3-ethanol, i3-acrylate, skatole, tryptamine), and high (indole, i3-acetamide, i3-pyruvate) efficacies. We displayed ligand-selective antagonist activities by i3-pyruvate, i3-aldehyde, indole, skatole, and tryptamine. Ligand binding assay identified low affinity (skatole, i3-pyruvate, and i3-acetamide) and very low affinity (i3-acrylate, i3-ethanol, indole) ligands of the murine AhR. Indole, skatole, tryptamine, i3-pyruvate, i3-acrylate, and i3-acetamide induced *CYP1A1* mRNA in intestinal LS180 and HT-29 cells, but not in the AhR-knockout HT-29 variant. We observed a similar CYP1A1 induction pattern in primary human hepatocytes. The most AhR-active catabolites (indole, skatole, tryptamine, i3-pyruvate, i3-acrylate, i3-acetamide) elicited nuclear translocation of the AhR, followed by a formation of AhR-ARNT heterodimer and enhanced binding of the AhR to the CYP1A1 gene promoter. Collectively, we comprehensively characterized the interactions of gut microbial tryptophan catabolites with the AhR, which may expand the current understanding of their potential roles in intestinal health and disease.

## 1. Introduction

The microbial community of the human gastrointestinal tract plays a vital role in the maintenance of gut health and host nutrition. The colonic bacteria also contribute to the regulation of functions of distant organs, including the brain, liver, immune system, and pancreas. The intestinal microbiome has also been linked to the etiology of inflammatory bowel disease, obesity, diabetes, or cardiovascular diseases [1]. Intestinal bacterial metabolites comprise a wide range of compounds; however, three major classes of compounds seem to play a significant role in intestinal physiology: short-chain fatty acids, secondary bile acids, and tryptophan catabolites. A plethora of microbial intestinal catabolites of tryptophan (MICT), including indole (IND), tryptamine (TA), skatole (3MI), indole-3-pyruvate (IPY), indole-3-lactate (ILA), indole-3-acrylate (IAC), indole-3-propionate (IPA), indole-3-acetamide (IAD), indole-3-acetate (IAA), indole-3-ethanol (IET), indole-3-aldehyde (IA), and indole-3-acetaldehyde, have been identified so far [2]. In recent years, it also became evident that a principal molecular target for indole-based compounds is the aryl hydrocarbon receptor (AhR). This receptor is a ligand-inducible transcription factor that resides in the cell cytosol in its resting state. Upon ligand binding, AhR translocates into the nucleus, where it heterodimerizes with AhR nuclear translocator (ARNT), and the AhR/ARNT heterodimer binds to the dioxin responsive elements within enhancer and promoter regions of the AhR target genes. The AhR plays multiple roles in the regulation of immune responses, xenoprotection, bone remodeling, carcinogenesis, cell regeneration, organ development, metabolic diseases, and neurophysiology [3,4,5,6,7]. The AhR ligands can be both xenobiotics and endogenous compounds [8,9,10]. A variety of indole-based compounds have been documented to act as AhR ligands, including indirubin and indigo [11], bilirubin, biliverdin and hemin [12], diindolylmethane [13] and other di-indole derivatives [14], indole-3-carbinol [15], indoxyl-3-sulfate [16,17], ultraviolet photoproducts of tryptophan [18], including 6-formylindolo[3 ,2-b]carbazole (FICZ) [19], marine brominated indoles [20], and several MICTs (TA, IAA [21], IA [17], IND [22], or 3MI [23,24]). Indoles may act as human AhR-selective agonists, but also as AhR antagonists in a context-specific manner [22,25].

Several isolated reports dealt with the biological effects of MICT via the AhR. Tryptophan microbial catabolite IA has been found to engage the AhR and to balance mucosal reactivity via interleukin-22 [26]. Lamas et al. have shown that deficiency in the microbial production of microbial tryptophan product IAA leads to increased susceptibility to colitis, involving the AhR-dependent pathways [27]. Therefore, it is of particular interest to know how individual MICT influence AhR signaling.

The intestinal concentrations of MICTs are mostly unknown. The most studied ones include the fecal levels of indole, where its concentrations have been reported to range from 250 up to 2500 µM [2,25,27,28]. The fecal concentrations of 3MI spanned from 40 to 750 µM in healthy subjects and patients with intestinal pathologies, respectively [29]. The concentration of IAA in fecal samples of healthy adults can approach five (5) µM [27]. We have found indole levels ranging from 40 to 130 µM in colonoscopy aspirates from 36 healthy and colitis human subjects, which suggests that the intestinal indole levels could be lower than those in feces. In the same cohort, we found IPA levels to be about 0.5 µM (unpublished observations, Mani Lab).

In the current study, we investigated in detail the effects of MICT on the AhR-CYP1A signaling pathway. We tested MICT at concentrations up to 200 µM (1 mM for indole) given their abundance in intestines. We found that MICT is low-affinity ligands of AhR, which act as low-potency and high-efficacy agonists of AhR. MICT induced nuclear translocation of AhR, the heterodimerization of AhR with ARNT, the binding of AhR to CYP1A1 promoter, and the induction of CYP1A1, in the AhR-dependent manner, in both hepatic and intestinal cells. The most efficacious MICT included IND, 3MI, TA, IPY, IAC, and IAD, whereas IAA, IPA, ILA, and IA were inactive.

## 2. Results

### 2.1. Tryptophan in the CULTURE Medium Does not Influence AhR Activity

We aimed to evaluate the biological activities of tryptophan metabolites. Since conventional cell culture media and fetal bovine serum contain Trp, which may impact the background AhR-mediated activity [30], we first validated the appropriateness of Trp-containing media for such a study. Using HPLC, we determined Trp concentration in cultures of AZ-AHR, LS174T, and HaCaT cells throughout 24 h. The initial concentration of Trp in culture medium containing 10% serum was approx. 75 µM, and it slowly linearly decreased, down to approx. 60 µM after 24 h of incubation, regardless of the cell-type being used (Appendix A). Therefore, we considered Trp concentration in cell cultures to be stable over the time of incubation. Trp did not activate AhR up to 100 µM concentration in AZ-AHR cell cultures when using Trp-free/serum-free medium, whereas both TCDD and IND induced luciferase activity by 10000-fold and 500-fold, respectively (Appendix A). We compared the activation of AhR by TDCC and IND in AZ-AHR cells cultured in conventional, Trp-free, serum-free, and Trp-free/serum-free media. We concluded that the presence of Trp in the culture medium would not interfere with MICT testing under standard cell culture conditions (Appendix A). Moreover, Trp did not displace ^3^H-TCDD from binding at the murine AhR (data not shown).

### 2.2. MICTs Exhibit Full and Partial Agonist Effects on AhR in the AZ-AHR Reporter Cell Line

In the first series of experiments, we evaluated the agonistic and antagonistic effects of MICT on the AhR activation. All compounds were tested at concentrations of up to 200 µM, except for IAC (the maximum concentration was 100 µM due to solubility limitations) and IND (10 mM, given its known high intestinal concentrations). All MICTs activated the AhR in a dose-dependent manner; however, we did not reach the plateau, and their relative efficacies differed substantially. The effectively inactive agonists were IPA, IAA, ILA, and IA (≈8-fold induction). 3MI, TA, IET, and IAC (fold inductions from 40-fold to 100-fold) displayed a medium efficacy agonism. In contrast, the most efficacious agonists included IND, IPY, and IAD (fold inductions from 200-fold to 600-fold). The relative potencies of MICT were rather low, with their EC_50_ ranging from 42 to 103 µM. A notable exception was IND, with EC_50_ ≈ 1.5 mM (Figure 1; upper panels). The antagonistic effects of MICT were evaluated against typical full agonists of the AhR, comprising TCDD, BaP, and FICZ, which were applied at fixed concentrations corresponding to their EC_80_. IA, IND, and TA displayed a dose-dependent antagonistic effect against all used agonists. In contrast, we observed no antagonistic effects against any of the full agonists to be exhibited by IET, IAC, ILA, IAA, and IPA. A ligand-selective antagonism was observed for 3MI, which dose-dependently inhibited the AhR activation by TCDD and BaP, but not by FICZ. Interestingly, IAD and IPY had dual effects on ligand-activated AhR. These two compounds antagonized TCDD, but potentiated the agonistic effects of BaP and FICZ (Figure 1; lower panels).

### 2.3. MICT Is Orthosteric Ligands of the AhR

The ability of MICT to bind the AhR was then evaluated by competitive radio-ligand binding assay, using cytosols from mouse hepatoma cells Hepa1c1c7. High-affinity AhR ligand FICZ actively displaced [^3^H]-TCDD from binding at AhR (IC_50_ ≈ 2 nM). Among tested MICT, we identified low-affinity AhR ligands including 3MI (IC_50_ 72 µM), IPY (IC_50_ 55 µM) and IAD (IC_50_ 44 µM), and very low-affinity ligands comprising IAC (IC_50_ 710 µM), IET (IC_50_ 1540 µM) and IND (IC_50_ 1130 µM) (Figure 2). Figure 3 shows a comprehensive chart and “heat map,” summarizing AhR binding affinities of individual MICT, as well as their respective agonistic and antagonistic effects towards the AhR.

### 2.4. MICT as Inducers of the AhR Target Gene CYP1A1

Since MICT acted as ligands and full/partial agonists of the AhR, we next examined their effects on the induction of expression of CYP1A1, a prototypical AhR target gene. We incubated intestinal and hepatic cells with MICT for 24 h and *measured CYP1A1* mRNA levels by qRT-PCR. We observed a strong induction of *CYP1A1* mRNA in intestinal LS180 cells by IND, IPY, 3MI, TA, IAC, and IAD (Figure 4A), which was largely consistent with the reporter gene assay data (except for IET). The antagonistic effects were observed primarily in the case of IND, 3MI, IPY, and IA, and to a lesser extent, also exhibited by IAA, IPA, ILA, and IAC, in LS180 cells co-incubated with TCDD (Figure 4B). This profile was only partially consistent with the reporter gene assay results, which could be due to the cell-type specific effects. A strong induction of *CYP1A1* mRNA was observed for IPY, IAC, and IAD, while IND, 3MI, and TA acted as weaker inducers in colon HT-29 cells. The qualitative profiles of *CYP1A1* mRNA induction in both intestinal cell models were identical. Moreover, CYP1A1 induction by TCDD and MICT was nullified in the AhR knock-out HT-29 variant (Figure 4C), which corroborates the involvement of AhR in MICT-dependent CYP1A1 induction. Finally, IND, IPY, 3MI, IAC, and IAD, but not TA, induced *CYP1A1* mRNA in primary cultures of human hepatocytes obtained from three different donors (Figure 4D). The lack of induction by TA could be, in part, related to its extensive hepatic metabolism.

### 2.5. The Lead MICT Trigger Nuclear Translocation of the AhR and the Formation of AhR-ARNT Heterodimer and Its Binging to the CYP1A1 Promoter

Lead AhR-active MICT, including IND, IPY, 3MI, IAC, IAD, and TA, were subjected to a series of assays for their ability to trigger molecular functions of AhR in LS180 cells. An early cellular event, following ligand binding, is the nuclear translocation of the AhR, where it forms a heterodimer with ARNT, which in turn binds DRE motifs in its target gene promoters. All tested lead MICT triggered a massive nuclear translocation of AhR, which was of a similar intensity to that that elicited by TCDD, as revealed by immune-fluorescence detection (Figure 5; Appendix A). Consequently, protein co-immune-precipitation assay confirmed that IND, IPY, 3MI, IAC, IAD, and TA induce a formation of AhR-ARNT heterodimer (Figure 6). Finally, the binding of the AhR to the promoter of the CYP1A1 gene was correspondingly enhanced by all lead MICT as by TCDD, as revealed by the results of the ChIP assay (Figure 7).

## 3. Discussion

In the current paper, we demonstrate that the microbial catabolite of tryptophan acts, to a different extent, as ligands and agonists of AhR. These actions of tryptophan catabolites are mainly supported by our observations that MICT: (i) displace [^3^H]-TCDD from the AhR in the ligand-binding assay; (ii) activate AhR in reporter gene assays; (iii) induce AhR target gene expression; (iv) trigger the nuclear translocation of AhR; (v) induce the formation of AhR-ARNT heterodimer; (vi) enhance the binding of the AhR to *CYP1A1* promoter. A resume of the obtained data is shown in Table 1.

Indeed, investigators have previously shown that several compounds formed in vivo and in vitro from Trp act as ligands and activators of AhR. These include photo-reactive Trp catabolites formed in the skin after irradiation by UV light [19], endogenous Trp metabolites [28], or microbial metabolites produced in the skin [31] or intestines [2,23]. Several (individual) Trp metabolites were studied in vitro for their AhR-mediated activities, but also in vivo, thus unveiling their potential physiological and pathological roles dependent on the AhR activation [17,26,27,31]. Nevertheless, a comprehensive study, describing the complex effects of the entire human or microbial Trp metabolome on the AhR has not been conducted so far. Our study is the first one to comprehensively describe the effects of all known human microbial intestinal catabolites of Trp on the AhR activity and functions.

We mainly focused on a comparative overview of effects occurring at the biologically relevant concentrations of MICT. For instance, Lamas et al. reported average fecal concentrations of IAA in healthy human subjects (*n* = 32) to be about five (5) µM, and a similar concentration of IAA (1 µM) has been found in mouse feces. However, they have observed a significant activation of the AhR (15-fold) by IAA only at concentrations exceeding 1000× those in vivo (approx. 5000 µM) [27]. Jin et al. reported IAA concentrations in mouse cecum and feces to range between 20 µM and 30 µM. They observed a very low induction of *CYP1A1* mRNA by 100 µM IAA (≈ 5-fold) in Caco-2 and MDA-MB-231 cells (but not in MDA-MB-468 cells); this induction was more robust (between 10-fold and 35-fold) with IAA in concentrations exceeding 500 µM [25]. Heath-Pagliuso et al. have also described the activation of mouse and guinea pig AhR by IAA, but only at a two (2)-mM concentration [21]. In the present study, we describe negligible activation of AhR (Figure 1) and no induction of *CYP1A1* mRNA in LS180 cells, HT-29 cells, and primary human hepatocytes (Figure 4) by 200 µM IAA. Overall, given the observable intestinal and fecal concentrations of IAA in human (≈5 µM), and the AhR-active concentrations of IAA in vitro (100–5000 µM), the potential AhR-mediated biological effects of IAA in vivo would be somewhat limited. Similarly, we found a negligible activation of AhR and no induction of *CYP1A1* mRNA by 200 µM IPA, which is consistent with data published by Hubbard et al. [22], who described no AhR activation by ten (10) µM IPA in HepG2 cells. Since the levels of IPA in colonoscopy aspirates from 36 healthy and colitis subjects can approach ≈0.5 µM (unpublished observations, Mani Lab), the biological effects of IPA mediated by the AhR could be negligible. The relevance of IND intestinal concentrations is critical for the assessment of its AhR-dependent effects in humans. The fecal levels of IND in human subjects range between 250 and 2500 µM [2,25,27,28]; however, its concentrations in colonoscopy aspirates ranged only from 40 to 130 µM (unpublished observations, Mani Lab). Similarly, fecal IND concentrations in mice were about 700 µM, while those in the cecum were about 300 µM. In the same study, there was a moderate induction of *CYP1A1* mRNA in various human cells at 500–1000 µM concentration of IND [25], which is consistent with data presented in our current study (Figure 1 and Figure 4). Interestingly, the activation of AhR and the induction of *CYP1A1* mRNA was achieved in HepG2 cells by 100 µM IND, but incubation time in this study was only four (4) h [22]. This observation corroborates our findings in the recent studies, where we observed a higher relative efficacy of methylindoles at AhR after four (4) h of incubation as compared to 24 h [24,32]. Fecal concentrations of 3MI vary substantially between healthy subjects (≈40 µM) and in patients with intestinal pathologies (≤750 µM) [29]. The activation of the AhR and induction of CYP1A1 by 3MI at concentrations occurring in intestines in vivo is reported here and in several other studies [22,23,24,32], which suggests that the AhR activation could contribute to known physiological and pathophysiological roles of 3MI in gut tissue. The intestinal concentration of TA in humans is largely unknown. Jin et al. found ten (10) and 15 µM TA levels in mouse feces and cecum, respectively, and observed *CYP1A1* mRNA induction by 50 µM TA in Caco-2 and MDA-MB-486/231 cells [25], which seems consistent with our findings in AZ-AHR, LS180 or HT-29 cells. Conflicting results exist for IA. We observed a weak affinity to the mouse AhR (IC_50_ > 1000 µM), as well as a very low efficacy (6-fold induction) and low potency (EC_50_ = 62 µM) towards the human AhR. Moreover, IA exhibited a weak antagonistic effect on the human AhR, when activated by TCDD, BaP, and FICZ (IC_50_ ≈ 100–200 µM). Previously, Zelante et al. reported a vigorous agonist activity of IA in mouse H1L1.1c2 reporter cells [26]. Yu et al. have demonstrated that IA attenuates inflammation in patients with atopic dermatitis through the AhR; however, direct evidence for IA effects on AhR has not been presented in that study [31].

The entire intestinal microbial metabolic pathway of Trp comprises twelve catabolites [2], of which we studied eleven compounds in detail. While the activity of several MICTs at AhR was reported (vide supra), five derivatives, including IAD, ILA, IPY, IAC, and IET, have not been previously evaluated. Importantly, except for the ILA, these MICTs appear to be the strongest AhR agonists and ligands within the Trp microbial catabolic pathway, which suggests the existence of underexplored chemical space in therapeutic targeting AhR, using microbial catabolites and their mimics. Interestingly, we have just recently described indole microbial metabolites-based mimics as selective ligands of human pregnane X receptor (PXR), with anti-inflammatory activity in vitro and in vivo [33]. AhR has also been implicated in the control of inflammation within the intestine; in particular, within its immune cell compartment [34]. Thus, more information about the impact of MICT (or their derivatives) on intestinal health is needed.

Finally, an intriguing future challenge is to investigate the mixed (combined) effects of MICT on the AhR in both mechanistic and translational studies. Since MICTs occur within intestines as complex mixtures, an understanding of their cumulative impact on the AhR might be a clue for the therapeutic targeting of AhR. Indeed, we have previously observed synergistic effects of several methylated and methoxylated indoles on the AhR in vitro [24]. Future studies should also address not only the combined effects of MICT themselves but their interactions with other classes of important intestinal microbial metabolites, including short-chain fatty acids and bile acids.

## 4. Materials and Methods

### 4.1. Chemicals and Reagents

Indole-3-aldehyde (97% purity), indole-3-ethanole (97% purity), benzo[a]pyrene (BaP; B1760, Lot SLBS0038V, purity 99%), 6-formylindolo[3,2-b]carbazole (FICZ; SML1489, Lot 0000026018, purity 99.5%), dimethylsulfoxide (DMSO), Triton X-100, bovine serum albumin, and hygromycin B were obtained from Sigma-Aldrich (Prague, Czech Republic). All other MICTs (purity ≥ 98% as determined by supplier) and anti-AhR (Alexa fluor 488) (SC-133088) antibody were purchased from Santa Cruz Biotechnology (Santa Cruz, CA, USA). 2,3,7,8-tetrachlorodibenzo-*p*-dioxin (TCDD) was from Ultra Scientific (RI, USA). 2,3,7,8-tetrachlorodibenzofuran (TCDF) was from Ambinter (Orleáns, France). Luciferase lysis buffer was from Promega (Madison, CA, USA). DAPI (4′,6-diamino-2-phenylindole) was from Serva (Heidelberg, Germany). [^3^H]-TCDD (purity 98.6%; ART 1642, Lot 181018) was purchased from American Radiolabeled Chemicals. Bio-Gel^®^ HTP Hydroxyapatite (1300420, Lot 64079675) was obtained from Bio-Rad Laboratories. All other chemicals were of the highest purity commercially available.

### 4.2. Cell Cultures

The stably transfected reporter gene AZ-AHR cell line was described previously [35]. Human Caucasian colon adenocarcinoma cell line LS180 (ECACC No. 87021202) and mouse hepatoma Hepa1c1c7 (ECACC No. 95090613) were purchased from the European Collection of Cell Cultures and cultured as recommended by the supplier.

CRISPR/Cas9 knockout of the AhR in HT-29 cells: A CRISPR/Cas9 expression vector with GFP, pSpCas9(BB)-2A-GFP (PX458) (#48138; Addgene, Cambridge, MA, USA) was used for integration of the gRNA complementary to exon 2 of the AHR gene. 4 × 10^5^ cells were seeded in a 6-well plate and transfected (at 60–70% confluence) with a mixture of vector DNA and Lipofectamine™ 3000 (Life Technologies) according to the manufacturer’s instructions. The transfected cells positive for GFP signal were sorted using BD Aria II Sorp (Becton Dickinson, Franklin Lakes, NJ, USA), in a single cell suspension into a 96-well plate. DNA isolation was performed by a QIAamp DNA Mini Kit (Qiagen, Hilden, Germany) according to the manufacturer’s instructions. The function of the CRISPR/Cas9 system was verified by the SURVEYOR mutation detection kit (Integrated DNA Technologies, Leuven, Belgium) according to the manufacturer’s instructions. The positive clones were checked for AhR expression by Western Blotting. PCR amplification products were purified using the QIAquick PCR Purification Kit (Qiagen) and integrated into a plasmid vector (pGEM_T Easy Vector Systems, Promega), and E.coli DHP α (MAX Efficiency DH5a Competent Cells, Thermo Fisher; Waltham, MA, USA) were transformed according to manufacturer’s instructions, DNA was isolated, and deletions in the AhR gene sequences were verified by sequencing (Mix2seq, Eurofins Genomic, Luxembourg). Appendix A shows the sequencing data of the HT-29 AhR KO clone E4 variant.

Primary human hepatocyte cultures were from two sources: (i) long-term human hepatocytes in monolayer batch Hep2201020 (male, 75 years, Caucasian) and Hep2201021 (male, 66 years, Caucasian) were purchased from Biopredic International (Rennes, France); (ii) primary human hepatocytes from multiorgan donor LH79 (male, 60 years, Caucasian) were prepared at Palacky University Olomouc. Liver tissue was obtained from Faculty Hospital Olomouc, and the tissue acquisition protocol followed the requirements issued by the “Ethical Committee of the Faculty Hospital Olomouc, Czech Republic” and Transplantation law #285/2002 Coll.

### 4.3. Reporter Gene Assay in the AZ-AHR Cells

AZ-AHR cells were incubated for 24 h with vehicle (DMSO; 0.1% *v*/*v*) and increasing concentrations of tested MICT in the presence or absence of TCDD (13.5 nM), FICZ (22.6 µM), or BaP (15.8 µM). Then, cells were lysed, and luciferase activity was measured using the Tecan Infinite M200 plate luminometer (Schoeller Instruments, Czech Republic). The experiments were performed in at least four consecutive cell passages, and the treatments were in quadruplicates (technical replicates). The values of half-maximal effective concentration (EC_50_) for each MICT, and half-maximal inhibitory concentration (IC_50_—where appropriate) were calculated.

### 4.4. mRNA Isolation and Quantitative Real-Time Reverse Transcriptase-Polymerase Chain Reaction (qRT-PCR)

Total RNA was isolated using TRI Reagent^®^ (Sigma Aldrich, Prague, Czech Republic). cDNA was synthesized using M-MuLV reverse transcriptase (New England Biolabs, Ipswich, MA, USA). The reverse transcription was performed at 42 °C for 60 min using Random Primers 6 (New England Biolabs, Ipswich, MA, USA) and diluted in a 1:4 ratio by PCR-grade water. The quantitative reverse transcriptase-polymerase chain reaction (qRT-PCR) was performed on the Lightcycler 480 II using the LightCycler ^®^ 480 Probes Master (Roche Diagnostic Corporation, Prague, Czech Republic). The levels of *CYP1A1* and *GAPDH* mRNAs were determined using the Universal Probe Library probes (UPL; Roche Diagnostic Corporation, Prague, Czech Republic) in combination with specific primers, using a protocol described elsewhere [36]. All measurements were performed in triplicates, and the gene expression was normalized to *glyceraldehyde-3-phosphate dehydrogenase (GAPDH)* as a house-keeping gene. The data were processed using the delta-delta Ct method and expressed as a fold induction over the negative control (DMSO) or as a percentage of the positive control (maximal induction by TCDD).

### 4.5. Competitive Radio-Ligand AhR Binding Assay

Ligand binding assay was performed using cytosolic protein extracts from murine Hepa1c1c7 cells as described [37]. Briefly, aliquots of protein (2 mg/mL) were incubated at the room temperature for two (2) h with 2 nM [^3^H]-TCDD in the presence of DMSO (vehicle control), FICZ (positive control), 200 nM TCDF (non-specific binding) or increasing concentration of MICT. After the incubation, the hydroxyapatite slurry was added to the samples and the suspension was incubated on ice and washed three times with HEGT buffer. The hydroxyapatite pellet was re-suspended in scintillation cocktail, and radioactivity was determined in a liquid scintillation counter. The specific binding of [^3^H]-TCDD was determined by subtracting the radioactivity of non-specific reaction (TCDF) from total radioactivity. The values of IC_50_ were calculated where appropriate.

### 4.6. Immunofluorescence Detection of the Nuclear Translocation of the AhR

LS180 cells (60,000 cells/well) were plated on 8-well chambered slides (94.6140.802, Sarstedt, Nümbrecht, Germany) and allowed to grow overnight. Then, the cells were incubated with DMSO (0.1% *V*/*V*), TCDD (10 nM), IND (1000 µM), IPY (200 µM), 3MI (200 µM), TA (200 µM), IAC (100 µM), and IAD (200 µM). The staining procedure was described in detail elsewhere [24]. The images were captured using an IX73 fluorescence microscope (Olympus, Tokyo, Japan). The immunofluorescence signal intensity (of the AhR antibody) in the nucleus was evaluated visually. For percentage calculation, approximately 500 cells from at least four randomly selected fields of view in each specimen were used.

### 4.7. Chromatin Immunoprecipitation Assay (ChIP)—the Detection of the Binding of the AhR to CYP1A1 Promoter

LS180 cells (4 million) were seeded in a 60-mm dish in Dulbecco’s modified Eagle’s medium (D6546; Sigma Aldrich). The following day, the cells were incubated with DMSO (0.1% *V*/*V*), TCDD (10 nM), IND (1000 µM), IPY (200 µM), 3MI (200 µM), TA (200 µM), IAC (100 µM), and IAD (200 µM) for 90 min at 37 °C. ChIP assay was performed as we described in detail elsewhere [24].

### 4.8. Protein Co-Immunoprecipitation—the Detection of the Formation of the AhR-ARNT Heterodimer

Intestinal LS180 cells were incubated with TCDD, tested MICT, and vehicle (DMSO; 0.1% *V*/*V*) for 90 min at 37 °C. The Pierce™ Co-Immunoprecipitation Kit (Thermo Fisher Scientific) with covalently coupled AhR antibody (mouse monoclonal, sc-133088, A-3, Santa Cruz Biotechnology) was used. Eluted protein complexes and total parental lysates were resolved on SDS-PAGE gels followed by Western Blot analysis and immuno-detection with ARNT-1 antibody (mouse monoclonal, sc-17812, G-3, Santa Cruz Biotechnology). Chemiluminescent detection was performed using horseradish peroxidase-conjugated anti-mouse secondary antibody (7076S, Cell Signaling Technology) and WesternSure^®^ PREMIUM Chemiluminescent Substrate (LI-COR Biotechnology) by C-DiGit^®^ Blot Scanner (LI-COR Biotechnology).

### 4.9. Determination of the Levels of Tryptophan in Cell Culture Media by High-Performance Liquid Chromatography (HPLC)

The levels of tryptophan were determined in the samples of culture media isolated from the growing cells in 2-h intervals from 0 h to 24 h. A simple gradient HPLC method using an Agilent 1260 Infinity system (consisting of a quaternary gradient pump, autosampler, column thermostat, and diode array detector (DAD)) was used. The starting mobile phase mix consisted of 90% of 0.2% trifluoroacetic acid (TFA) and 10% of acetonitrile (MeCN), and, during the run, a linear gradient was applied for up to 15 min, increasing the percentage of MeCN to 90%. A post time of 5 min was applied to achieve the proper starting conditions for the next run. The stable mobile phase flow of 0.5 mL/min was applied at the reverse phase chromatographic column Agilent Poroshell 120 EC-C18, 4.6 × 50 mm, 2.7 μm particle size. The DAD detector was set to detect the following wavelengths: 290 ± 4 nm (A), 254 ± 4 nm (B), 210 ± 4 nm (C), and 275 ± 4 nm (D) (used for the quantification of the tryptophan levels) with a reference range of wavelengths of 820 ± 100 nm. The sample volume applied to the column was ten (10) μL in all cases. The external calibration in the concentration range of 0.7–112.0 µM was used, giving a straight calibration curve with R2 = 0.9999. The peak of tryptophan was identified at t_R_ = 5.0 ± 0.1 min, and in all determinations was completely resolved from other peaks in the chromatogram. The individual determinations were done in duplicate, and the concentration levels were expressed as mean values ± sample standard deviations.

### 4.10. Human Studies

The colonoscopy aspirate or fecal collection study was collected under Institutional Review Board (IRB)-approved studies for aspirate or fecal collections (#2015-4465; #2009-446; #2007-554). The patients eligible for colonoscopy were enrolled sequentially after they provided study consent (#2015-4465; audited by the IRB on 24 April 2019). All patients were screened and evaluated by a single gastroenterologist and Inflammatory Bowel Disease specialist (DL). Patients were enrolled if they had a diagnosis of inflammatory bowel disease (Crohn’s disease or Ulcerative colitis) or were undergoing routine screening colonoscopy for colorectal polyps/cancer screening or required a colonoscopy as part of their medical management of any gastrointestinal disorder as clinically indicated. IBD and control tissue pathology were obtained under a separate protocol (CCI#2007-554).

### 4.11. Statistical Analyses

In order to determine significantly different results over the negative control (vehicle; DMSO; 0.1%), one-way analysis of variance (ANOVA), followed by Dunnett’s test, was applied. The result was considered significant if the p-value was lower than 0.05. All the calculations (including IC_50_ and EC_50_) were performed using GraphPad Prism version 8.0 for Windows (GraphPad Software, La Jolla, CA, USA).

## 5. Conclusions

A number of recent studies have indicated that the impact of MICTs on the AhR activity contributes to the control of intestinal homeostasis in both health and disease conditions. In this study, we provide a comprehensive quantitative characterization of the effects of microbial Trp metabolome on the human AhR, using a series of intestinal and liver cell models. We particularly focused on the effects of MICTs, which would occur at their biologically relevant concentrations, based both on literary evidence and our own results of analyses of colonoscopy aspirates. Our data suggest that all of the studied MICTs may act as low potency agonists of the AhR; however, their action on the AhR may differ substantially, as they can, in turn, behave as full agonists, partial agonists, or even ligand-selective antagonists of the human AhR. However, for some compounds, these effects may occur well beyond their intestinal levels. The activity of MICT in the AhR reporter gene assay mostly corresponded with their binding to the receptor, induction of its nuclear translocation, and binding to DNA within the CYP1A1 regulatory regions, as well as CYP1A1 transcription. Importantly, for the first time, we show that IAD, IPY, IAC, and IET, four MICTs that have not been previously evaluated, appear to be among the strongest AhR ligands within the microbial Trp metabolic pathway. Given the proposed role(s) of the AhR in the control of the gut immune system or intestinal inflammation, it will be important to address the role(s) of these MICT both individually or as complete mixtures in future mechanistic and translational studies addressing gut health.

## Figures and Tables

**Figure 1 ijms-21-02614-f001:**
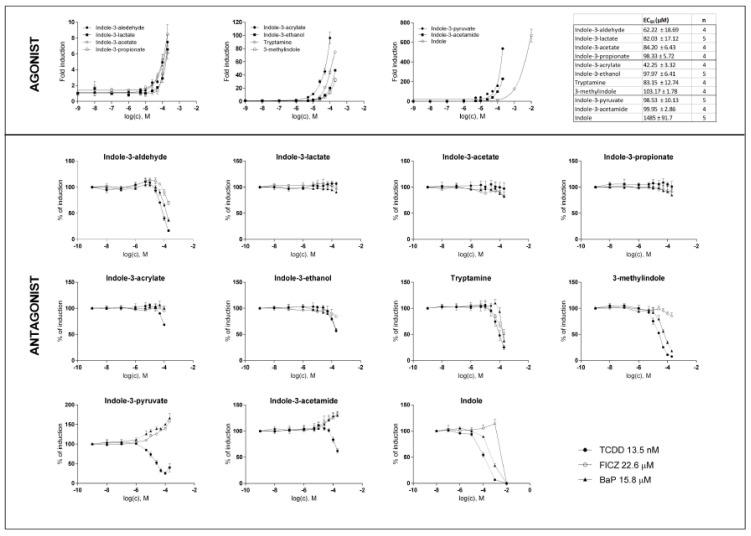
Effects of MICT on the AhR transcriptional activity in the reporter gene assay. AZ-AHR cells were incubated for 24 h with vehicle (DMSO; 0.1% v/v) and increasing concentrations of MICT, in the absence (agonist analyses) or the presence (antagonist analyses) of TCDD (13.5 nM), BaP (15.8 µM) and FICZ (22.6 µM). Following the treatments, we lysed cells, and luciferase activity was measured. Experiments were performed at least in four consecutive cell passages. Incubations were performed in quadruplicates (technical replicates). In agonist analyses, the data from a representative experiment, expressed as a fold induction of luciferase activity over control cells, are shown. In antagonist analyses, the data are a percentage of maximal induction, and they are the mean ± SD. The inserted table shows the number of cell passages and calculated EC_50_ for each test compound.

**Figure 2 ijms-21-02614-f002:**
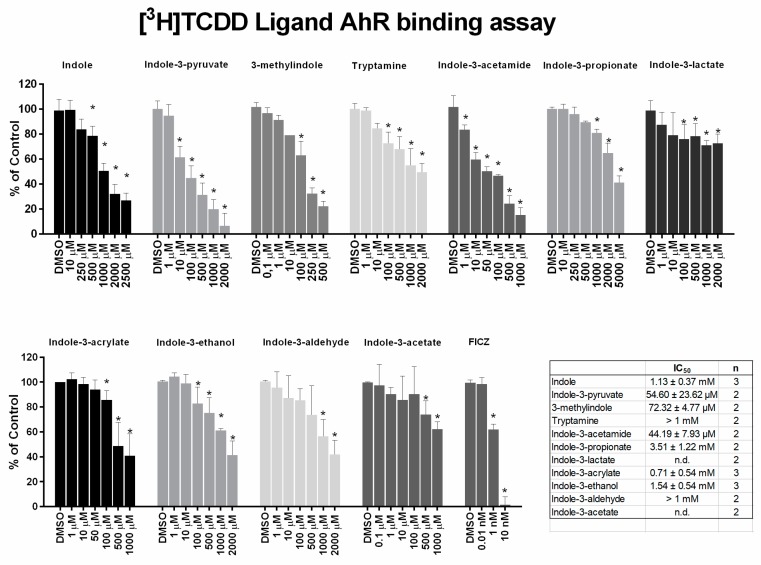
Competitive radio-ligand binding assay. Cytosols from Hepa1c1c7 cells were incubated with vehicle (negative control; 0.1% *V/V*), FICZ (positive control; 0.01 nM – 10 nM), TCDF (non-specific binding; 200 nM) and increasing doses of MICT in the presence of 2 nM [^3^H]-TCDD. The specific binding of [^3^H]-TCDD was determined as a difference between total and non-specific (200 nM; TCDF) reactions (value for vehicle DMSO; 0.1% *V*/*V* = corresponds to *specific binding of [^3^H]-TCDD = 100%*). At least two independent experiments were performed and the incubations were done in triplicates in each experiment (technical replicates). The error bars represent the mean ± SD. * = significantly different from the vehicle (*p* < 0.05). The inserted table shows the number of repeats and IC_50_ values for each MICT.

**Figure 3 ijms-21-02614-f003:**
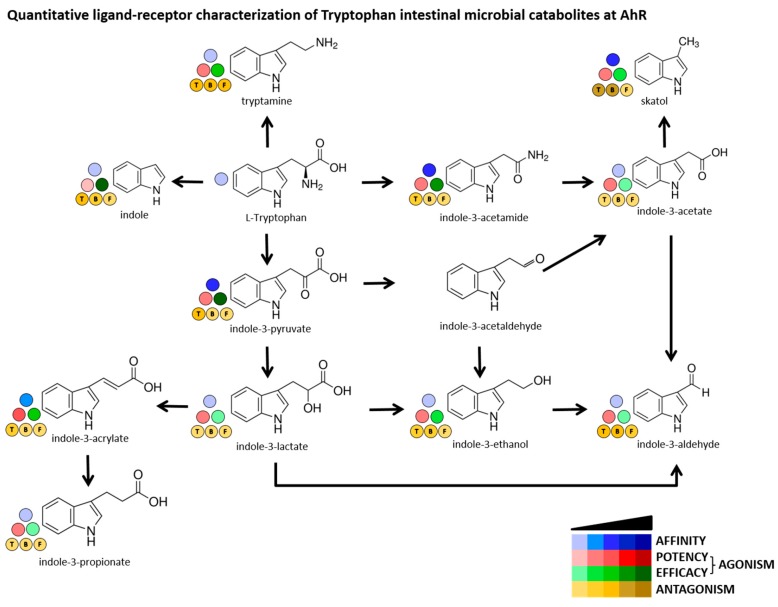
Quantitative characterization of interactions between MICT and AhR. A scheme depicts intestinal microbial catabolism of tryptophan, and the source data come from Figure 1 and Figure 2. The Blue scale refers to the affinity of MICT (ligand binding). The Red & Green scales quantify the relative agonist effects of MICT; Red ≈ potency (EC_50_), Green ≈ efficacy (E_MAX_). The Brown scale quantifies the relative antagonist effects (IC_50_) of MICT against three different agonists used at EC_80_ concentration and designated as “T” = TCDD, “B” = BaP, “F” = FICZ.

**Figure 4 ijms-21-02614-f004:**
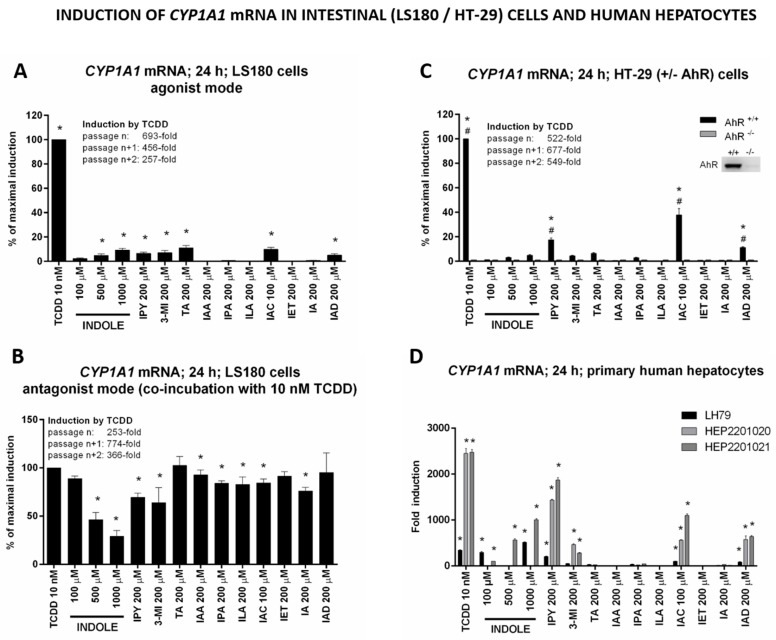
Effects of MICT on the induction of *CYP1A1* mRNA in human hepatic and intestinal cells. Cultured cells were incubated for 24 h with the vehicle (DMSO; 0.1% *v/v*) and tested MICT in the presence or absence of TCDD (10 nM). RT-PCR determined the level of *CYP1A1* mRNA, and the data were normalized per *GAPDH* mRNA level. Each measurement was done in triplicates (technical replicates). * = a value significantly different from the negative control (*p* < 0.05). (**A**,**B**) Experiments in three consecutive passages of human colon adenocarcinoma cell line LS180 in the absence (**A**) and the presence (**B**) of 10 nM TCDD. The bar graphs show the percentage of maximal induction attained by TCDD and are expressed as the mean ± SD. (**C)** Experiments in three consecutive passages of wild-type (AhR^+^/^+^) and AhR-knockout (AhR^−^/^−^) HT-29 cells. The bar graph shows a percentage of maximal induction achieved by TCDD. The data are expressed as the mean ± SD. # = a value significantly different from HT-29 wild-type (AhR^+^/^+^) (*p* < 0.05). Insert Western Blot shows confirmation of AhR knock-out. (**D**) Primary human hepatocytes from three different donors (LH79, Hep2201020, and Hep2201021). The bar graph shows a fold induction of *CYP1A1* mRNA over vehicle-incubated cells.

**Figure 5 ijms-21-02614-f005:**
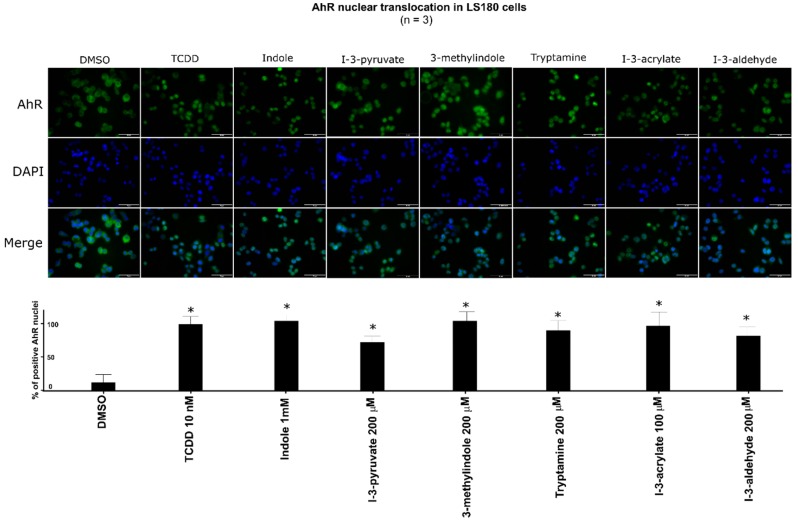
Effect of MICT on nuclear translocation of the AhR. LS180 cells were incubated for 90 min with vehicle (DMSO; 0.1% *V*/*V*), TCDD (10 nM), and test MICT. The experiments were performed in three consecutive cell passages with all tested compounds in duplication. Fluorescence images depict sub-cellular localization of AhR (upper panels) and nuclear staining by DAPI (lower panels). Representative micrographs are shown. The bar graph shows the percentage of AhR positive nuclei (mean ± SD; *n* = 3) relative to TCDD-treated cells (also ref. Appendix A). * = a value significantly different from negative control (*p* < 0.05).

**Figure 6 ijms-21-02614-f006:**
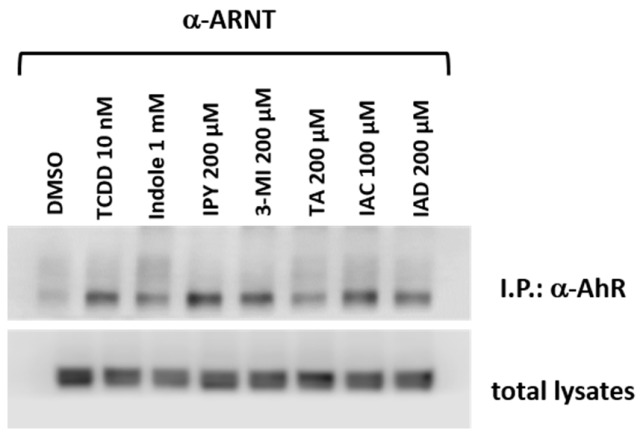
MICTs induce a formation of the AhR-ARNT heterodimer. Intestinal LS180 cells were incubated with TCDD, tested MICT, and vehicle (DMSO; 0.1% *V*/*V*) for 90 min. Protein co-immunoprecipitation of ARNT was carried out, as described in the Methods section. The representative immunoblots of immuno-precipitated protein eluates and total cell lysates are shown. The experiments were performed in two consecutive cell passages.

**Figure 7 ijms-21-02614-f007:**
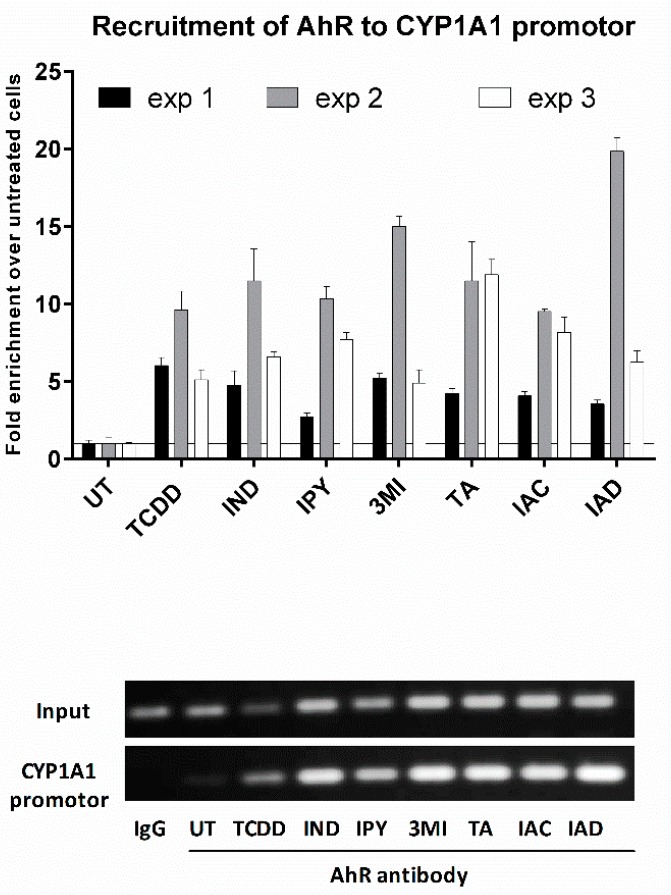
MICT elicit the binding of the AhR to *CYP1A1* promoter, as determined by the chromatin immunoprecipitation assay (ChIP). LS180 cells were incubated with vehicle, TCDD, and tested compounds, as described in Materials and Methods, and were then subjected to ChIP analysis. Bar graphs show binding of the AhR to CYP1A1 promoter as quantified by RT-PCR. The entire protocol was carried out in three consecutive cell passages. The data are expressed as a fold enrichment to vehicle-treated cells and show the mean ± SD from duplicates (technical replicates). DNA fragments amplified by PCR were resolved on 2% agarose gel (representative record from “exp 2” is shown).

**Table 1 ijms-21-02614-t001:** Summary of MICT effects on the AhR–CYP1A1 pathway.

Compound	Affinity (IC_50_)	Potency (EC_50_)	Efficacy (E_MAX_)	Antagonism (IC_50_)	Gene Expression	AhR Cell Functions (Translocation; Heterodimerization; DNA-Binding
**Indole**	very low	very low	high	yes–all ligands	strong inducer	highly active–all parameters
**Skatole**	low	low	medium	ligand selective	strong inducer	highly active–all parameters
**Tryptamine**	no	low	medium	yes–all ligands	strong inducer	highly active–all parameters
**I3-acetamide**	low	low	high	none	strong inducer	highly active–all parameters
**I3-acetate**	no	low	low	none	inactive	not tested
**I3-acrylate**	very low	low	medium	none	strong inducer	highly active–all parameters
**I3-aldehyde**	no	low	low	yes–all ligands	inactive	not tested
**I3-ethanol**	very low	low	medium	ligand selective	inactive	not tested
**I3-lactate**	no	low	low	none	inactive	not tested
**I3-propionate**	no	low	low	none	inactive	not tested
**I3-pyruvate**	low	low	high	ligand selective	strong inducer	highly active–all parameters

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
