# Peer review of "Gut Microbial Catabolites of Tryptophan Are Ligands and Agonists of the Aryl Hydrocarbon Receptor: A Detailed Characterization"

_ijms, 2020, doi:10.3390/ijms21072614_

Round 1

Reviewer 1 Report

I must to congratulate for your excellent work. It is really great. There are few grammar and typographical errors easy to fix. Please ask for a native Engish scientist to improve your manuscript. 

Author Response

REVIEWER 1

I must to congratulate for your excellent work. It is really great.

Comment 1/1: There are few grammar and typographical errors easy to fix. Please ask for a native Engish scientist to improve your manuscript.

Response 1/1: We thank reviewer for complimentary and encouraging opinion. English language editing was done by a co-author who is native speaker. In addition, manuscript was screened through PubSURE platform.

Reviewer 2 Report

In the paper entitled “Gut microbial catabolites of tryptophan are ligands and agonists of aryl hydrocarbon receptor: a detailed characterization” Barbora Vyhlídalová and co-workers, the authors reported on the effect of gut microbial catabolites of tryptophan on AhR activity.

The manuscript is clearly a very interesting topic in the current research. As a matter of fact, accumulating evidences show that microbial tryptophan catabolites activate the immune system through binding to AhR.

These interactions contribute to intestinal and systemic homeostasis in health and disease conditions.

-In my opinion the abstract is confusing; it must be rewritten clarifying, first of all, research aims and then the results in a more concise manner.

-In the Introduction, Page 2 line 58 the authors must add some references when talk about xenobiotics capable of activating AhR.

-Fig. 1 is not well readable, especially the first raw of graphs. I suggest organizing the data in a clearer way; consider also the possibility to put part of the data in the supporting information file.

-Figure 5, the legend text (under the histograms) is too small and so difficult to read.

-The discussion is clear and well written; anyway a table which resumes the results could be useful.

-Conclusions Section is missing. I suggest adding it, considering the amount of data reported and discussed in the manuscript it is very important to give to the readers the final take home highlights of the scientific findings.

Author Response

REVIEWER 2

In the paper entitled “Gut microbial catabolites of tryptophan are ligands and agonists of aryl hydrocarbon receptor: a detailed characterization” Barbora Vyhlídalová and co-workers, the authors reported on the effect of gut microbial catabolites of tryptophan on AhR activity.

The manuscript is clearly a very interesting topic in the current research. As a matter of fact, accumulating evidences show that microbial tryptophan catabolites activate the immune system through binding to AhR. These interactions contribute to intestinal and systemic homeostasis in health and disease conditions.

Comment 2/1: In my opinion the abstract is confusing; it must be rewritten clarifying, first of all, research aims and then the results in a more concise manner.

Response 2/1: An abstract was re-written as suggested by the reviewer.

Comment 2/2: In the Introduction, Page 2 line 58 the authors must add some references when talk about xenobiotics capable of activating AhR.

Response 2/2: We included two extra references: Denison&Nagy 2003 Ann Rev Pharmacol Toxicol 43:309; Abel&Haarmann-Stemmann 2010 Biol Chem 391:1235.

Comment 2/3: Fig. 1 is not well readable, especially the first raw of graphs. I suggest organizing the data in a clearer way; consider also the possibility to put part of the data in the supporting information file.

Response 2/3: We redesigned Figure 1 in a way to get it more clear and comprehensible. Given the amount and structure of the data contained in the figure, the transfer of some data in supplement would be rather contra-productive in terms of clarity.

Comment 2/4: Figure 5, the legend text (under the histograms) is too small and so difficult to read.

Response 2/4: The text legend under histogram was enlarged in the revised manuscript.

Comment 2/5: The discussion is clear and well written; anyway a table which resumes the results could be useful.

Response 2/5: A summary table was elaborated and inserted in Discussion section of the revised manuscript.

Comment 2/6: Conclusions Section is missing. I suggest adding it, considering the amount of data reported and discussed in the manuscript it is very important to give to the readers the final take home highlights of the scientific findings.

Response 2/6: We thank reviewer for this suggestion. We included new section “conclusions” in the revised manuscript.
